# Cvill6 and Cvill7: Potent and Selective Peptide Blockers of Kv1.2 Ion Channel Isolated from Mexican Scorpion *Centruroides villegasi*

**DOI:** 10.3390/toxins17060279

**Published:** 2025-06-04

**Authors:** Kashmala Shakeel, Muhammad Umair Naseem, Timoteo Olamendi-Portugal, Fernando Z. Zamudio, Lourival Domingos Possani, Gyorgy Panyi

**Affiliations:** 1Department of Biophysics and Cell Biology, Faculty of Medicine, Research Center for Molecular Medicine, University of Debrecen, Egyetem ter. 1, 4032 Debrecen, Hungary; shakeel.kashmala@med.unideb.hu (K.S.); umair.naseem@med.unideb.hu (M.U.N.); 2Departamento de Medicina Molecular y Bioprocesos, Instituto de Biotecnologia, Universidad Nacional Autónoma de México, Av. Universidad 2001, Cuernavaca 62210, Mexico; timoteo.olamendi@ibt.unam.mx (T.O.-P.); fernando.zamudio@ibt.unam.mx (F.Z.Z.)

**Keywords:** α-KTx, *Centruroides villegasi* peptides, Kv1.2, Kv1.3, KCa3.1, patch-clampelectrophysiology, scorpion toxin

## Abstract

Scorpion venoms are a rich source of peptides that modulate the activity of ion channels and can serve as a new drug for channelopathies. Cvill6 and Cvill7 are two new peptides isolated from the venom of *Centruroides villegasi* with MW of 4277 Da and 4287 Da and they consist of 38 and 39 amino acids, respectively, including six cysteines. Sequence alignment revealed high similarity with members of the α-KTx2 subfamily of potassium channel toxins. In electrophysiology, Cvill7 potently inhibited Kv1.2 ion channels with an IC_50_ of 16 pM and Kv1.3 with an IC_50_ of 7.2 nM. In addition, it exhibited partial activity on KCa3.1 and Kv1.1, with ~16% and ~34% inhibition at 100 nM, respectively. In contrast, Cvill6 blocked Kv1.2 with low affinity (IC_50_ of 3.9 nM) and showed modest inhibition of Kv1.3 (~11%) and KCa3.1 (~27%) at 100 nM concentration. Neither peptide showed any activity against other K^+^ channels tested in this study (Kv1.5, Kv11.1, KCa1.1, and KCa2.2). Notably, Cvill7 has a remarkable affinity for Kv1.2 and high selectivity of 450-fold over Kv1.3 and 12,000-fold over Kv1.1. These pharmacological properties make Cvill7 a potential candidate to target Kv1.2 gain of function (GOF)-related channelopathies such as epilepsy.

## 1. Introduction

Ion channels are present in nearly all organisms, with potassium channels being the most widely distributed [1]. Voltage-gated potassium (Kv) channels represent the largest family of K^+^ channels and are encoded by 40 distinct genes, classified into 12 subfamilies. The Kv1 subfamily (Kv1.1–Kv1.8) is abundantly prevalent in the nervous, vascular, and immune systems [2,3]. Voltage-gated K^+^ channels play a crucial role in regulating essential physiological functions, including maintaining membrane potential, regulating cell volume, modulating calcium signaling, controlling cell proliferation, and initiating action potentials in both excitable and non-excitable cells [4,5,6]. Thus, due to the diverse functions and patterns of expression of K^+^ ion channels their dysfunctions have been linked to numerous channelopathies [7,8,9].

The Kv1.2 ion channel, a member of voltage-gated potassium channels, is encoded by the *KCNA2* gene. Kv1.2 is predominantly expressed in the central nervous system (CNS) abundantly in the cerebellum, hippocampus, and cortex [10,11,12]. It plays a crucial role in action potential propagation, neuronal excitability, and synaptic transmission [13]. Gain-of-function (GOF) mutations in *KCNA2* cause increased Kv1.2 activity, resulting in excessive potassium conductance leading to neuronal hyperpolarization, and impairing normal brain function [14]. A decade ago, Kv1.2 emerged as a key factor in neurologic excitability disorders when Syrbe et al. identified gain-of-function mutations R297Q and L298F in the *KCNA2* gene of patients exhibiting early-onset epileptic encephalopathy (EE), intellectual impairment, motor delay, and ataxia [15]. Subsequently, several other GOF mutations in the *KCNA2* gene, such as E157K, H310Y, H310R, and E236K, were identified and proven as major contributors to Kv1.2-related channelopathies [16,17,18,19,20]. Generally, Kv1.2 GOF mutations exhibited a 9- to 13-fold increase in current amplitude, accompanied by a −50 mV shift in the Kv1.2 activation threshold, making Kv1.2 hyperactive [15]. Recently, studies have shown that blocking Kv1.2 currents with 4-aminopyridine (4-AP), a small organic compound, can mitigate the gain-of-function defects in patients with *KCNA2*-related encephalopathy [21,22]. Currently, only symptomatic treatments, such as antiepileptic drugs, are available to treat epileptic encephalopathy [23]. Therefore, a potent and specific inhibitor of Kv1.2 would be helpful in restoring the normal neuronal excitability and alleviating the pathological effects associated with Kv1.2 GOF mutations in EE patients.

A promising approach to achieve selective Kv1.2 inhibition is to investigate natural peptide toxins and their analogues. Venomous animals, particularly scorpions, are a rich source of natural unique peptides that inhibit ion channels with varying potency and selectivity [24,25]. Until now more than 218 scorpion toxins, classified into seven families (α-Ktx, β-Ktx, γ-Ktx, δ-Ktx, ε-Ktx, κ-Ktx, and λ-Ktx), have been identified that target K^+^ channels [26,27,28]. The α-KTx toxins (potassium channel scorpion toxins) share structural and functional similarities, but only few target selectively a specific member among the Kv1 channel family (Kv1.1–Kv1.8) [29]. Achieving selective binding is complicated, as multiple interaction points between the toxin and channel residues determine the specificity for an ion channel [30,31]. In addition, Kv1 channels share significant sequence homology and structural similarities, particularly in the pore domain where toxins bind [32]. Therefore, most α-KTxs inhibit more than one Kv1 subtype. For example, Margatoxin and Hongotoxin-1 have similar affinities for both Kv1.3 and Kv1.2 [33,34,35,36]. Only a few toxins have been isolated from scorpions which specifically target Kv1.2 over Kv1.1 and Kv1.3. Therefore, searching for a highly selective and potent blocker of Kv1.2 which can be used to treat Kv1.2-related channelopathies motivates us to further explore the venom of various scorpions.

Mexico is a rich country in scorpion species, containing circa 12% of the world biodiversity [37,38,39,40]. The species dangerous to man are very wide, being over 20 different species [41]. This is in part due to the geographic localization (tropical and sub-tropical areas) with a substantial variation in weather, temperature, rain, and type of vegetation, from forest sites to desertic areas. We are presently very interested in studying the species that cause human problems, hoping to develop a new type of antivenom [40]. It is known that the different species contain a variety of toxic peptides that recognize voltage-dependent ion channels, mainly sodium (Na^+^) channels, but contain also a great variability of compounds that recognize potassium channels. Initially, we described toxins from the species *Centruroides villegasi* that affect Na^+^-channels [37,40]; however, we know that the presence of toxins that target potassium channels is also widely distributed in species of the genus Centruroides. In this study, we report the isolation and characterization of two novel peptides Cvill6 and Cvill7 from *C. villegasi* venom. The primary structures of these peptides were determined using Edman degradation and compared with those of known potassium scorpion toxins (KTxs). To assess their pharmacological activity and selectivity, the peptides were tested against eight different voltage-gated or Ca^2+^-activated K^+^ ion channels using single-cell electrophysiology (patch-clamp) assays.

## 2. Results

### 2.1. Purification and Primary Sequence Determination of Cvill Peptides

A comprehensive detail of the purification and identification of new toxins from the *Centruroides villegasi* venom which target the Na^+^ channel was described in our previous publication [40], and here we repeat the procedure but focus on K^+^ channel toxins. In brief, a three-step purification procedure was chosen to obtain pure peptide toxins from the venom. In the first purification step, soluble venom from *C. villegasi* was separated using a Sephadex G-50 column, resulting in three different fractions: I, II, and III (Appendix A). Since only fraction II showed toxicity to mice, as described in our previous report [40], and this fraction is also known to contain peptides that inhibit ion channels [42,43], it was further purified to isolate peptides by subjecting it to a carboxymethylcellulose (CMC) column, yielding 12 distinct subfractions (Appendix A). Subsequently, selective subfractions (F-II-8 to F-II-11 which were toxic to mice as reported previously [40]) were purified by high-performance liquid chromatography (HPLC) using a C_18_ column, and subfractions F-II-10 and F-II-11 held two new toxins, specific for K^+^-channels (Figure 1A,B). Peaks eluted at 25.5 min in the chromatograph of the F-II-10 subfraction (Figure 1A, marked with a green asterisk) and at 23.6 min in the chromatograph of F-II-11 (Figure 1B, indicated with a red asterisk) contained pure peptides, as confirmed by HPLC using a C_18_ column (Appendix A) and MS analysis (Appendix A). The F-II-10 25.5 toxin with the molecular weight (MW) of 4277.03 Da (Appendix A) and the F-II-11 23.6 toxin with MW of 4287.12 Da (Appendix A) are named Cvill6 and Cvill7, respectively (according to a recent publication on the trivial names of scorpion toxins) [44]. The primary structure of the pure peptides as shown in Figure 2 was determined by automated Edman degradation, following the same procedure used for other components from the same venom; first, a pure native peptide was sequenced, followed by sequencing a reduced and alkylated sample to identify cysteine residues [40]. The peptide sequences were submitted to UniProt Knowledgebase under the accession numbers C0HME3 for Cvill6 and C0HME4 for Cvill7.

### 2.2. Comparative Sequence Analysis and Prediction of Structural Features of Cvill Peptides

A search for peptides similar in sequence to Cvill peptides was conducted using BLASTP, followed by the alignment with MAFFT (version 7). The analysis revealed that both Cvill6 and Cvill7 share significant sequence similarities with members of the α-KTx 2 subfamily of scorpion K^+^-channel blockers (Figure 2A,B). Cvill6 exhibited 97% identity with toxin Ce3 (α-KTx 2.10), 92% with CboK6 (α-KTx 2.21), and 92% with CIITx1 (α-KTx 2.3). In contrast, Cvill7 showed 95% identity with toxin Ce1 (α-KTx 2.8), toxin Ce2 (α-KTx 2.9), and Ct28 (α-KTx 2.20). Considering this high identity index, Cvill6 and Cvill7 were classified as α-KTx 2.25 and α-KTx 2.26, respectively, and the newest members of the α-KTx 2 subfamily.

Structural modeling of the Cvill6 and Cvill7 peptides (Appendix A) using the AlphaFold3 tool [45] showed that both Cvill peptides have a single α-helix connected to an anti-parallel β-sheet stabilized by three disulfide bonds (CS α/β). In general, a similar motif has been described in the other known structures of α-KTx toxins. To demonstrate this, we performed structural alignment of the predicted models of the Cvill peptides and NMR solution structure of MgTx (MgTx, α-KTx 2.2, Appendix A) as shown in Appendix A.

### 2.3. Effect of Cvill Peptides on Voltage-Gated Potassium Channels

The primary amino acid sequences of Cvill6 and Cvill7 exhibit a high degree of identity to members of the α-KTx 2 subfamily, which are generally known as blockers of voltage-gated K^+^ ion channels, particularly Kv1.1, Kv1.2, and Kv1.3. Therefore, we aimed to investigate the effects of Cvill6 and Cvill7 on voltage-gated K^+^ ion channels. Among the tested ion channels there were four members from the Shaker-related human Kv1 family, hKv1.1, hKv1.2, hKv1.3, and hKv1.5 (Figure 3A–E), and one member of the human ether-a-go-go-related (hERG) cardiac K^+^ channel hKv11.1 (hERG1, Figure 3F). For current recordings, Kv1.1, Kv1.2, and Kv1.5 were transiently expressed in CHO cells while for Kv11.1 a stable HEK293 cell line was used. To record Kv1.3 currents human peripheral T lymphocytes were activated with Phytohemagglutinin A (PHA) to boost Kv1.3 expression and Ca^2+^-free intracellular solution to prevent the KCa3.1 channel opening. Hence, whole-cell currents were measured exclusively from Kv1.3 channels. Appropriate voltage protocols were applied to record the K^+^ currents from voltage-clamped cells in whole-cell configurations. Toxins were freshly dissolved in the bath solution supplemented with 0.1 mg/mL BSA and applied to the patched cells via a gravity-fed micro-perfusion system with an approximate flow rate of 200 µL/min. The complete exchange of solutions in the bath chamber and the proper functioning of the perfusion system were regularly verified using fully reversible inhibitors as positive controls, such as TEA^+^ for Kv1.1, Kv1.3, and Kv1.5; charybdotoxin (ChTx) for Kv1.2; and E4031 1µM for Kv11.1. The inhibition of peak current upon perfusing the cell with these positive controls as shown in Figure 3A–F (traces in blue) verified the expression of the proper ion channel and indicated the characteristics (kinetics and completeness) of the solution exchange in the recording chamber.

Figure 3A–F shows the representative whole-cell K^+^ current traces for various Kv channels, recorded sequentially in the same cell for the respective ion channel, in the absence of toxin (control, black trace) and in the presence of Cvill6 (green trace) or Cvill7 (red trace) toxin at a concentration of 1 nM (only for Kv1.2) or 100 nM. Traces in the presence of the toxins were recorded upon reaching the equilibrium block or after ~3 min of perfusion with toxin solution when no apparent block was observed. Representative electrophysiology records show that Cvill6 inhibits ~28% of hKv1.2 current at 1 nM concentration (Figure 3B) but only ~13% of the hKv1.3 current is inhibited at 100 nM concentration (Figure 3D). However, no change in current amplitudes was observed for hKv1.1 (Figure 3A), hKv1.5 (Figure 3E), and hKv11.1 (Figure 3F) at 100 nM of Cvill6. On the other hand, Cvill7 blocked >97% of the hKv1.2 current at 1 nM concentration (Figure 3C), indicating an extraordinary potency. In addition, Cvill7 also affected the hKv1.1 (Figure 3A) and Kv1.3 (Figure 3D) currents by reducing ~32% and ~86% of the peak currents at equilibrium block with 100 nM concentration, respectively. Nevertheless, Cvill7 did not show any effect on hKv1.5 (Figure 3E) and hKv11.1 (Figure 3F) currents. Figure 3G shows a summary of toxin activity on Kv channels indicating that Cvill6 affects Kv1.2 and Kv1.3 while Cvill7 affects Kv1.1, Kv1.2, and Kv1.3. The estimated IC_50_ values of Cvill6 from a single concentration, based on the bimolecular interaction of channel and toxin (1:1) [46], yielded ~0.84 µM for Kv1.3. Similar estimates for Cvill7 yielded ~192 nM for Kv1.1. The quantity of native toxins was insufficient to conduct the full concentration–response analysis when the affinity of the toxin for a channel was low, above 50 nM.

### 2.4. Cvill7 Selectively Inhibits Kv1.2 over Kv1.3 with Low-Picomolar Affinity

Since Cvill7 peptide demonstrated remarkably high affinity towards hKv1.2 in our initial screening against Kv channels, we further extended our experiments to determine the concentration-dependent inhibition of Kv1.2 and Kv1.3 currents by Cvill7 and its binding kinetics. The development and recovery of the block of Kv1.2 at 100 pM concentration of Cvill7 are shown in Figure 4A. Normalized peak currents (I_norm_ = I_t_/I_0_, where I_t_ is the peak current at time t and I_0_ is the peak current in the absence of toxin at t =0) for individual cells (*n* = 4) were averaged and plotted against the time. The loss of Kv1.2 current apparently saturated to 96 ± 1.4% of the initial (t = 0) peak current in 7–8 min upon application of 100 pM of Cvill7. Fitting the individual block kinetics (cell-by-cell) using a single-exponential decay function (see Section 5 for details) gave the time constant (τ_on_) of 104 ± 15 s (*n* = 4) for the development of the block (Appendix A). The dissociation kinetics of Cvill7 were also extremely slow. The equilibrium block was followed by the application of toxin-free solution for ~15 min, during which just one-fifth of the blocked current recovered with extremely slow kinetics, suggesting that Cvill7 is a virtually irreversible inhibitor of Kv1.2. The dissociation constant (τ_off_) of Cvill7 for the Kv1.2 channel was 5978 ± 538 s (*n* = 4), obtained by fitting a single-exponential rising function to the normalized peak currents during the wash-out procedure (see Section 5.4.4 for details, Appendix A). On the other hand, unlike Kv1.2, the block of Kv1.3 by Cvill7 was fully reversible with rapid association and dissociation kinetics. The onset of the equilibrium block to 67 ± 3% occurred with a τ_on_ of 20 ± 1.8 s (*n* = 3) upon 15 nM of toxin exposure and it fully recovered to the initial normalized peak current with a τ_off_ of 34 ± 6 s (*n* = 3) upon perfusing the cell with toxin-free solution, as shown in Figure 4C and Appendix A. The kinetic parameters of toxin binding, association rate constant (k_on_), and dissociation rate constant (k_off_) of Cvill7, as given in Table 1, were also calculated from the toxin wash-in and wash-out time constants (τ_on_ and τ_off_) for Kv1.2 and Kv1.3 channels assuming the simple bimolecular interaction between the toxin and the channel [35,47], as previously described for ChTx binding to the Shaker K^+^ channel [46].

To determine the concentration-dependence of the current inhibition of Kv1.2 and Kv1.3, the equilibrium block was determined at different concentrations of Cvill7 peptide. Considering the extremely slow blocking kinetics of Cvill7 for Kv1.2, we applied the toxin for a sufficient period to reach the complete apparent equilibrium block. At low-picomolar concentrations the concentration–response curve was determined in a cumulative manner. The remaining current fractions (RCFs) were determined using the ratio I/I_0_, where I_0_ represents the peak current in the absence of the toxin, and I corresponds to the peak current at the equilibrium block in the presence of the toxin at a known concentration and plotted as a function of toxin concentration. The concentration–response curves were obtained by fitting the data points with the Hill equation (refer to Section 5 for details). The best fit resulted in the IC_50_ value of 16 ± 1.56 pM with a Hill coefficient (H) of 1.03 for Kv1.2 (Figure 4B) and IC_50_ value of 7.2 ± 0.64 nM with an H coefficient of 0.73 for Kv1.3 (Figure 4D). Thus, Cvill7 displays ~450-fold selectivity for Kv1.2 over Kv1.3.

### 2.5. Cvill6 Inhibits Kv1.2 with Nanomolar Affinity

Cvill6 peptide has 74% identity with Cvill7. However, during the screening experiments for Kv channels, it moderately inhibited hKv1.2 currents (~16% at a 1 nM concentration) and displayed slight inhibition of Kv1.3 (~11% at a 100 nM concentration) (Figure 3G). To further understand the binding features of Cvill6 with Kv1.2, we conducted a similar set of experiments as for Cvill7. The onset and relief from the block of Kv1.2 by Cvill6 are shown in Figure 5A and time constants were determined as described in Section 2.4 and shown in Appendix A. Upon application of 100 nM of Cvill6, it rapidly attained the steady-state block to 97 ± 1.2% with the τ_on_ of 35 ± 2.7 s (*n* = 5). Unlike Cvill7, the block of Kv1.2 by Cvill6 was reversible but with very slow dissociation kinetics, and recovery up to 65% of the initial current occurred with the τ_off_ of 1287 ± 259 s (*n* = 5) following the application of the toxin-free solution. The binding kinetics of Cvill6 to Kv1.2 channels were also calculated based on their bimolecular interaction (see Section 2.4 for details) from the time constants and resulting rate constants are given in Table 1.The concentration-dependence of the block was obtained as detailed in Figure 5. Fitting of the Hill equation to mean RCF values at different concentrations of Cvill6 resulted in an IC_50_ of 3.9 ± 0.27 nM and Hill coefficient of 1.4 (Figure 5B).

### 2.6. Activity of Cvill6 and Cvill7 Toxins on Ca^2+^-Activated Potassium Channels

The scorpion toxins from the α-KTx 2 family also modulate the function of Ca^2+^-activated potassium channels, in addition to Kv1 channels [48,49]. Therefore, we investigated the activity of Cvill6 and Cvill7 toxins on three different types of Ca^2+^-activated K^+^ channels: (1) hKCa2.2 (SK2), the small-conductance Ca^2+^-activated channel; (2) hKCa3.1 (IKCa1, SK4), the intermediate-conductance Ca^2+^-activated channels, mainly expressed in T lymphocytes; and (3) mKCa1.1 (BK or MaxiK), the large-conductance voltage- and Ca^2+^- activated channel of mice. Appropriate depolarization protocols were applied to record the K^+^ current through respective KCa channels as shown in the inset of each panel in Figure 6A–C. The representative current traces, shown in Figure 6A–C, were sequentially recorded from the same cell expressing the respective KCa channel before the toxin application (black), in the presence of toxin for 2–3 min, or at equilibrium block at 100 nM concentration of Cvill6 (green) or Cvill7 (red). Traces in blue represent the equilibrium block of the respective channels in the presence of the positive control (TEA^+^ for KCa1.1, apamin for KCa2.2, and Cm39 toxin [50] for KCa3.1), confirming both the ion channel and the proper operation of the perfusion system. Neither Cvill6 nor Cvill7 showed any effect on KCa1.1 or KCa2.2 (Figure 6A,B,D). Cvill6 reduced ~40% of the KCa3.1 current (Figure 6C), where the RCF value was 0.73 ± 0.05 (*n* = 5) at 100 nM (Figure 6D). On the other hand, Cvill7 inhibited ~25% of the current at the same concentration (Figure 6C), with an RCF value 0.84 ± 0.03 (*n* = 5) (Figure 6D). The time course of inhibition of KCa3.1 currents at 100 nM of Cvill6 or Cvill7 is shown in Figure 6E and Figure 6F, respectively. The normalized peak currents (see Section 2.4 or Section 5 for details) for individual cells (*n* = 5) were averaged at each time point and plotted as a function of time. The steady-state block was reached in 5–6 consecutive traces (~1 min) after perfusing the cell with 100 nM of Cvill6 or Cvill7 toxins. The recovery from the block with Cvill6 to the initial peak current occurred in 4–5 consecutive traces (~0.8 min) and in the case of Cvill7 it took only 2–3 consecutive traces (~0.5 min) after perfusing the cell with toxin-lacking solution. This demonstrates that both Cvill6 and Cvill7 inhibit the KCa3.1 channel reversibly with quick association and dissociation kinetics. The estimated IC_50_ values for KCa3.1 from a single concentration (see above) yielded ~268 nM and ~527 nM of Cvill6 and Cvill7, respectively. Thus, Cvill6 has a two-fold higher affinity for KCa3.1 than Cvill7. The quantity of native toxins was insufficient to conduct the full dose–response curves at this high concentration range (see above).

## 3. Discussion

Peptide toxins purified from venomous species, such as scorpions, have gained significant interest due to their potential to modulate the ion channel function, thereby providing the opportunity to manage channelopathies. In this study, we described the purification, primary structure determination, and electrophysiological characterization of two new peptide toxins isolated from the venom of Mexican scorpion *Centruroides villegasi*. Cvill6 and Cvill7 consist of 38 and 39 amino acids with six cysteines, respectively. In electrophysiology assays, Cvill6 inhibited Kv1.2 with nanomolar affinity (IC_50_ = 3.9 nM) and blocked Kv1.3 and KCa3.1 currents by ~11% and ~27% at 100 nM concentration. On the contrary, Cvill7 inhibited Kv1.2 with an IC_50_ of 16 pM, exhibiting a remarkably high potency, and inhibited Kv1.3 with an IC_50_ of 7.2 nM. In addition, Cvill7 also inhibited ~34% and ~16% of Kv1.1 and KCa3.1 currents at 100 nM. However, several other voltage-gated (Kv1.5 and Kv11.1) or Ca^2+^-activated (KCa1.1 and KCa2.2) channels tested in this study remained insensitive to high concentrations of Cvill6 or Cvill7 peptides.

The primary amino acid sequence analysis revealed that Cvill6 and Cvill7 have a large extent of similarity with the already known 24 members of the α-KTx 2 subfamily of scorpion toxins, having all the six cystine residues conserved. Moreover, Lys28 and Tyr37, a “functional dyad” which is generally considered essential for a high-affinity block of Kv1 ion channels, is also present in both peptides [51]. Cvill6 has more than 90% of identity with scorpion toxins Ce3 (α-KTx 2.10), CboK6 (α-KTx 2.21), and CIITx1 (α-KTx 2.3) [52,53,54]. On the other hand, Cvill7 has more than 90% identity with toxins Ce1 (α-KTx 2.8), Ce2 (α-KTx 2.9), Ct28 (α-KTx 2.20), and CboK4 (α-KTx 2.23) [52,54,55]. Based on the high identity percentages, Cvill6 and Cvill7 were considered as the new members of the α-KTx 2 subfamily with systematic names (α-KTx 2.25 and α-KTx 2.26, respectively). It has already been proven that members of the α-KTx 2 subfamily are moderate- to high-affinity blockers of Kv1 ion channels, especially Kv1.2 and Kv1.3. In line with this, testing the activity of Cvill toxins revealed that Cvill6 has moderate affinity for Kv1.2 (IC_50_ = 3.9 nM) and is slightly active for Kv1.3 (IC_50_ = ~0.84 µM), exhibiting ~215-fold selectivity for Kv1.2 over Kv1.3. Toxin Ce3, a close relative of Cvill6 which differs only by a single residue of Asp to Ser at position 8, inhibits Kv1.2 and Kv1.3 with 376 pM and 126 nM IC_50_ values, respectively, and has ~335-fold selectivity for Kv1.2 [52,54]. This 10-fold decrease in affinity of Cvill6 occurred due to a change in a single amino acid. On the other hand, Cvill7 inhibited Kv1.2 with low-picomolar affinity (IC_50_ = 16 pM) and Kv1.3 with low-nanomolar affinity (IC_50_ = 7.2 nM), having a 450-fold selectivity for Kv1.2 over Kv1.3. The closely related toxins of Cvill7, namely Ce1, Ce2, and Ct28 (>94% identical), have not been evaluated for Kv1.2 inhibition according to available literature. For Kv1.3, Ce1 and Ce2 have 0.7 nM and 0.25 nM IC_50_ values. We previously reported CboK4 that inhibits Kv1.2 (IC_50_ = 125 pM) and Kv1.3 (IC_50_ = 22 nM) with ~176-fold selectivity for Kv1.2 over Kv1.3. It differs by three residues (~92% similarity) from Cvill7, which resulted in only an eight-fold decrease in the affinity of CboK4 for Kv1.2 [52]. The affinity of toxins for Kv1.2 and Kv1.3 is largely influenced by the presence of the “functional dyad” consisting of a conserved Lys residue and an aromatic or polar residue ~6–7 Å apart. This latter residue of the dyad is Tyr for high-affinity blockers of Kv1.2 and Thr or Asn for high-affinity blockers of Kv1.3 [27,51,56]. Comparison of amino acid sequences revealed that Cvill6 shares 74% identity with Cvill7, and both have the Lys28–Tyr37 dyad; however, Cvill6 has 244 times less potency for Kv1.2 than Cvill7 with the IC_50_s in low nanomolar and picomolar ranges, respectively. Based on these results, we conclude that the other differences in amino acids between Cvill6 and Cvill7 may correspond to the determinants of high affinity for Kv1.2 which can be further explored through molecular dynamics studies of toxin and channel interaction.

For quantitative insight into the binding kinetics of Cvill peptides to Kv1 channels, we determined the association and dissociation rate constants (k_on_ and k_off_, as shown in Table 1). The dissociation constant (K_d_) values, calculated by taking the k_off_/k_on_ ratios, of Cvill 7 are 1.7 pM and 25 nM for Kv1.2 and Kv1.3, respectively, and in the case of Cvill6, it is 3.3 nM for Kv1.2. These calculated K_d_ values are comparable to the IC_50_ values obtained from the concentration–response relationship (as given in Figure 4 and Figure 5); similar findings were previously reported for many other Kv1 channel blockers such as Cm39, Cm28, ChTx, and MgTx [33,46,50,57]. The differences may arise from the difficulty in the determination of the block equilibrium, especially at low toxin cointegrations, the incomplete wash-out of the toxin, and the very long experiments that can be influenced by the run-down of the current. Due to these limitations, we only determined the k_on_ and k_off_ values for one concentration of the peptide for each channel: 100 pM for Cvill7 and Kv1.2; 15 nM for Cvill7; and Kv1.3 and 100 nM for Cvill6 and Kv1.2. Considering the difficulties in obtaining precise time and rate constants, we preferred to report the IC_50_ values and gave the K_d_ values as supporting evidence. It is also worth noting that the fitting of the concentration–response relationships using the Hill equation resulted in Hill coefficients of close to 1 (between 0.7 and 1.4) and wash-in and wash-out kinetics were well fit using single-exponential functions (see Appendix A). These data indicate that, as with many of the pore blockers, Cvill7 and Cvill6 are pore blockers that interact with channels with 1:1 stoichiometry in a bimolecular interaction [46,57]. The presence of Lys residue, shown to be important to plug the pore of K^+^ channels [51,56,58], of the functional dyad in both Cvill6 and Cvill7 further supports the pore-blocking property of these peptides.

Several venom-derived toxins are known to date that block the Kv1.2 channel with high affinity, for example Margatoxin (MgTx, α-KTx 2.2), Hongotoxin-1 (α-KTx 2.5), and Pi1 (α-KTx 6.1), but these also inhibit other members of Kv1 channels like Kv1.3 with similar potencies [31,33,34,35,36,59]. Finding selective inhibition among Kv1 ion channel members, especially Kv1.1, Kv1.2, and Kv1.3, is a major challenge in Kv1 channel pharmacology, as they have high similarities in the pore region of channels where mostly scorpion toxins bind, thereby explaining the poor selectivity profile of toxins for Kv1.2 and Kv1.3 [32,60,61,62]. Despite these, subtle differences may exist among Kv1 channel types in the extracellular vestibule (generally called the “turret” and “selectivity filter”) which can contribute to the differential affinities for scorpion toxins [58,63,64]. There are only two known scorpion toxins so far which have remarkably high affinity for Kv1.2 and high selectivity over Kv1.3. Pi4 (α-KTx 6.4), isolated from the scorpion *Pandinus imperator*, has 8 pM IC_50_ for Kv1.2 and does not inhibit Kv1.3 at 10 µM concentration, having > 1 million times selectivity for Kv1.2. However, Pi4 inhibits KCa2 channels with an IC_50_ of 500 nM [65,66]. We previously reported Cbok7 (α-KTx 2.24), a peptide toxin purified from *Centruroides bonito*, which has 850-fold selectivity for Kv1.2 (IC_50_ of 24 pM) over Kv1.3 (IC_50_ of 20.4 nM) [52].

Cvill7, as demonstrated in this study, has an outstandingly high affinity for Kv1.2 (IC_50_ value of 16 pM) and significant selectivity over Kv1.3 (450-fold), Kv1.1 (~12,000-fold), and KCa3.1 (>32,000-fold). It showed no inhibitory effects on cardiac ion channels Kv1.5 and Kv11.1, and other Ca^2+^-activated channels (KCa1.1 and KCa2.2) tested in this study, thereby highlighting its preferential specificity for Kv1.2 and suggesting it as a strong therapeutic candidate. Cvill7 may be exploited to treat channelopathies associated with gain-of-function mutations of Kv1.2 such as severe epileptic encephalopathy (EE) [15,19,21]. The selective inhibition of mutated Kv1.2 channels on neurons in the central nervous system (CNS) using Cvill7 as a potential drug will have a minimum risk of off-target effects on other K^+^ channels (Kv1.1, Kv1.3, Kv1.4 and K1.6) which are also widely expressed in CNS [13,67]. However, the delivery of Cvill7 across the blood–brain barrier (BBB) will be challenging. This can be overcome by conjugating the Cvill7 with BBB shuttle peptides which have been previously shown to significantly improve CNS penetration and enhance peptide bioavailability [68,69,70].

## 4. Conclusions

In conclusion, we characterized two new peptides from the venom of Mexican scorpion *Centruroides villegasi*, named Cvill6 and Cvill7, which belong to the α-KTx 2 family. Cvill6 showed low nanomolar affinity for Kv1.2 and also inhibited KCa3.1 whereas Cvill7 emerged as a potent blocker of Kv1.2 having an IC_50_ value in the low picomolar range with 450-fold selectivity over Kv1.3. Cvill7 also demonstrated three orders of magnitude selectivity over several other ion channels. The exceptional potency and selectivity of Cvill7 make it an excellent candidate to target gain-of-function-associated Kv1.2 channelopathies.

## 5. Materials and Methods

### 5.1. Toxin Purification and Primary Sequence Determination

The venom was extracted from scorpions via electric stimulation, dissolved in sterile water, and centrifuged at 15,000 rpm at 4 °C for 15 min. The supernatant was collected, lyophilized, and stored at −20 °C. The detailed experimental procedure to isolate the peptides from the venom of *Centruroides villegasia* has been described previously [40]. Briefly, 33 mg of soluble venom was fractionated using a Sephadex G-50 column into F-I, -II, and -III fractions and the F-II was further purified by a carboxymethylcellulose (CMC) column. Out of 12 subfractions from the CMC column, the toxic fractions were separated by HPLC on the C_18_ column. The mass spectrometry analysis of pure peptides was performed with an ESI QTOF-MS instrument (maXis II UHR ESI-QTOF MS, Bruker, Bremen, Germany) and the amino acid sequencing was performed by Edman degradation using a PPSQ-31A/33A machine obtained from Shimadzu Protein Sequencer (Columbia, MD, USA).

### 5.2. Alignment of Primary Sequences with Known Toxins

To find the phylogenetically related toxins with Cvill6 and Cvill7, BLASTP (https://blast.ncbi.nlm.nih.gov/Blast.cgi, accessed on 12 February 2025) was performed. Multiple sequence alignment of mature chains was performed by MAFTT version 7 (https://mafft.cbrc.jp/alignment/server/, accessed on 12 February 2025) [71,72].

### 5.3. Modeling of Cvill Peptides

The tertiary structural prediction of peptides was performed using AlphaFold3, an AI-based tool designed for predicting protein structures. Based on the amino acid sequences of peptides AlphaFold3 generated multiple folding models. The model with the highest reliability scores was chosen, which had a pTM of 0.72 and a pLDDT score (the per-atom confidence measure) exceeding 90 [45].

### 5.4. Electrophysiology

#### 5.4.1. Cells and Heterologous Expression of Ion Channels

Chinese hamster ovary (CHO) cells (generously provided by Yosef Yarden, Weizmann Institute of Science, Rehovot, Israel) or human embryonic kidney (HEK293) cells were cultured and maintained in Dulbecco’s modified Eagle medium (DMEM, Thermo Fisher Scientific, Waltham, MA, USA) supplemented with 2 mM L-glutamine, 10% fetal bovine serum (FBS), 100 µg/mL streptomycin, and 100 U/mL penicillin-g (Sigma-Aldrich, Budapest, Hungary) following standard culturing conditions [52]. For transient expression of ion channels, CHO cells were transfected using the Lipofectamine 2000 kit (Invitrogen, Waltham, MA, USA) following the manufacturer’s instructions with plasmids encoding specific ion channels, including hKv1.1 (*KCNA1* gene), hKv1.2 (*KCNA2* gene), and hKv1.5 (*KCNA5* gene) in a pCMV6-AC-GFP vector (OriGene Technologies), hKCa2.2 (*KCNN2* gene) in a pCDN3 plasmid (for this transfection a GFP containing plasmid was co-transfected, courtesy of Bernard Attali, Tel Aviv University, Tel Aviv, Israel), and hKCa3.1 (*KCNN4* gene) in a pEGFP-C1 vector (a kind gift by H. Wulff, University of California, Davis, CA, USA). Cells expressing GFP were identified using a Nikon TE 2000U fluorescence microscope (Nikon, Tokyo, Japan) with bandpass filters set to 455–495 nm for excitation and 515–555 nm for emission. Electrophysiological recordings were typically conducted 24 to 48 h after transfection.

HEK 293 cells stably expressing mKCa1.1 1 (BKCa, *Kcnma1* gene, generously provided by C. Beeton from Baylor College of Medicine, Houston, TX, USA) or hKv11.1 (hERG1, *KCNH2* gene, gifted by H. Wulff from University of California, Davis, CA, USA) were utilized in this study.

For Kv1.3 currents recordings, peripheral blood mononuclear cells (PBMCs) were isolated from the healthy donor’s blood using the Histopaque 1077 separation method (Sigma-Aldrich, Budapest, Hungary) following approval from the Ethical Committee of the Hungarian Medical Research Council (36255-6/2017/EKU). The isolated PBMCs were activated for 3–6 days to boost Kv1.3 expression using Phytohemagglutinin A (PHA) (Sigma-Aldrich, Budapest, Hungary) in RPMI 1640 medium (Gibco, Grand Island, NY, USA) containing 10% FBS (Sigma-Aldrich, Budapest, Hungary), 2 mM L-glutamine, 100 µg/mL streptomycin, and 100 µg/mL penicillin as described previously [50].

#### 5.4.2. Solutions

The extracellular solution (bath solution) for both Kv and KCa channels consisted of 145 mM NaCl, 5 mM KCl, 2.5 mM CaCl_2_, 1 mM MgCl_2_, 10 mM HEPES, and 5.5 mM glucose, pH 7.35, and osmolarity between 302 and 308 mOsM/L. For positive control solutions containing various concentrations of tetraethylammoniums (TEA^+^), Na^+^ was substituted with TEA-Cl in equimolar concentration in the bath solution and all other ingredients remained unchanged. To avoid toxin adsorption onto the plastic surfaces of the perfusion system, 0.1 mg/mL of bovine serum albumin (BSA, Sigma-Aldrich, Budapest, Hungary) was also added into all bath solutions prior to the patch-clamp assay. The internal solution (pipette filling solution) for Kv1.1, Kv1.2, Kv1.3, Kv1.5, and mKCa1.1 was composed of 140 mM KF, 2 mM MgCl_2_, 1 mM CaCl_2_, 11 mM EGTA, and 10 mM HEPES, having a pH of 7.22, and for Kv11.1 it consisted of 140 mM KCl, 2 mM MgCl_2_, 10 mM EGTA, and 10 mM HEPES, pH of 7.22. For KCa2.2 and KCa3.1 the composition of the internal solution was 150 mM K-Aspartate, 5 mM HEPES, 8.5 mM CaCl_2_, 2 mM MgCl_2_, and 10 mM EGTA, pH 7.22, with an estimated free Ca^2+^ of ∼1−2 μM based on the MaxChelator program WEBMAX-C software (C. Patton, Stanford University: https://somapp.ucdmc.ucdavis.edu/pharmacology/bers/maxchelator/webmaxc/webmaxcE.htm, accessed on 18 July 2024). The osmolarity of the internal solutions was ~295 mOsM/L.

All the chemicals and salts for the control test solutions (TEA-Cl) were purchased from Sigma-Aldrich, Budapest, Hungary. Apamin and charybdotoxin were acquired from Smartox Biotechnology (Saint Egrève, France) and Cm39 was chemically synthesized in-house [50].

#### 5.4.3. Current Recording Conditions

Whole-cell currents were recorded using the patch-clamp technique in voltage-clamp mode, following standard protocols [73]. Data acquisition was carried out with a Multiclamp 200B amplifier connected to a personal computer via an Axon Digidata 1440 digitizer and Clampex 10.7 software (Molecular Devices, Sunnyvale, CA, USA). Current traces were typically low-pass-filtered with the amplifier’s built-in analog 4-pole Bessel filters and sampled at 4–50 kHz, ensuring the sampling rate was at least twice the filter cutoff frequency. Micropipettes were made from GC150F-7.5 borosilicate capillaries (Harvard Apparatus Co., Holliston, MA, USA) using a Sutter P2000 laser puller with tip resistance generally ranging from 3–6 MΩ in the bath solution. Only recordings with a leak current at the holding potential (V_h_) of <10% of the peak current at the test potential were included in the data analysis. All recordings were performed at room temperature (20–25 °C). Control and test solutions were perfused to the cells via a gravity-driven micro-perfusion system and AutoMate Perfusion Pencil Multi-Barrel Manifold Tip (AutoMate Scientific, Berkeley, CA, USA), with an approximate flow rate of 200 µL/min, and the excess amount of bath solution was continuously removed from the recording chamber using vacuum suction.

To evoke the K^+^ currents from the Kv1.x channels, depolarization pulses to +50 mV from a holding potential (V_h_) of −120 mV were applied every 15 s. The duration of depolarization pulses was 15 ms for Kv1.3 and 50 ms for Kv1.1 and Kv1.5. However, for Kv1.2, 15–500 ms-long pulses were used to achieve the saturated peak current due to the variable activation kinetics of this channel [74]. For Kv11.1 channels, currents were elicited by applying a voltage step to +20 mV for 1.25 s from a V_h_ of −80 mV followed by a step to −40 mV for 2 s, during which peak currents were measured, with pulses delivered every 30 s. mKCa1.1 currents were recorded by applying 600 ms-long voltage steps to +100 mV from a V_h_ of −100 mV. KCa2.2 and KCa3.1 currents were recorded by applying 150 ms-long voltage ramps to +50 mV from −120 mV every 10 s; the V_h_ was set to −85 mV.

#### 5.4.4. Data Analysis and Statistics

The Clampfit 10.7 software package (Molecular Devices, Sunnyvale, CA, USA) was used to analyze current recordings. Before analysis, all current traces were digitally filtered using three-point boxcar smoothing and were corrected for ohmic leakage, only if required. The blocking effect of the toxin at a given concentration was calculated as remaining current fraction (RCF = I/I_0_), where I represents the peak current at the equilibrium block at a given toxin concentration or peak current recorded after 3 min perfusion of toxin solution in the absence of a measurable block, and I_0_ is the peak current in the absence of the toxin. Data points in the concentration–response curve represent the mean RCF values of three to six individual cells, with error bars denoting the standard error of the mean (SEM). The data points for each concentration were fitted using the Hill equation: RCF = IC_50_^H^/(IC_50_^H^ + [toxin]^H^), where IC_50_ is the dissociation constant, H represents the Hill coefficient, and [toxin] is the concentration of the toxin.

To study the binding kinetics of Cvill toxins, the peak current during the application of toxin at time point t (I_t_) was normalized to the peak current (I_0_) before the toxin exposure (I_norm_ = I_t_/I_0_) and plotted as a function of time. The association time constant (*τ*_on_) was determined by fitting a single-exponential function to the data points during the toxin wash-in procedure (Equation (1), one-phase decay to RCF), and for the dissociation time constant (*τ*_off_), a single-exponential function rising to maximum was fitted to the data points during the wash-out procedure (Equation (2), RCF followed by one-phase association) for individual cells [35,57].(1)$$I_{norm}\left(t\right)=RCF+\left(\left(1-RCF\right)\times e^{-\frac{t}{\tau_{on}}}\right)\;$$(2)$$I_{norm}\left(t\right)=RCF+\left(\left(1-RCF\right)\times \left(1-e^{-\frac{t}{\tau_{off}}}\right)\right)\;$$

The time constants (*τ*_on_ and *τ*_off_) were used to calculate the association rate constant (k_on_) and dissociation rate constant (k_off_) based on a simple bimolecular interaction between the channel and the toxin, using Equations (3) and Equation (4), respectively [46,47,50].(3)$$k_{on}=\frac{1-\left(\tau_{on}\times k_{off}\right)}{\tau_{on}\times \left[toxin\right]}\;$$(4)$$k_{off}=\frac{1}{\tau_{off}}\;$$

Representative graph plotting and statistical analysis were conducted using the GraphPad Prism software package (version 8.0.1, La Jolla, CA, USA). All the data were presented with standard errors of the mean (SEM).

## Figures and Tables

**Figure 1 toxins-17-00279-f001:**
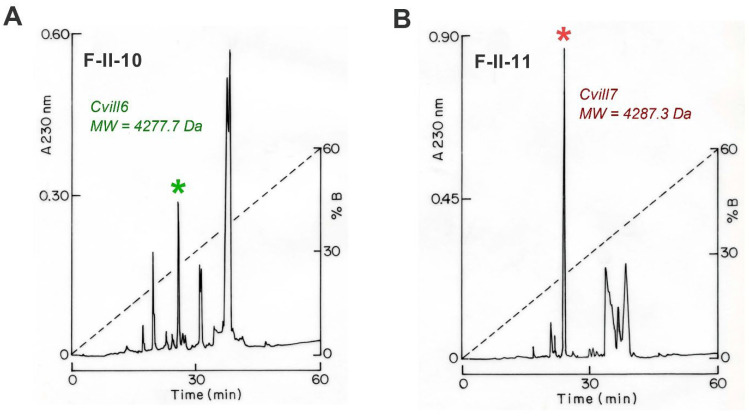
Isolation of native Cvill6 and Cvill7 from *Centruroides villegasi* venom [40]. The final step in the purification process, HPLC purification of fractions F-II-10 (**A**) and F-II-11 (**B**), was carried out using a C_18_ column (see Section 5 for details). Peptides were separated using a linear gradient, starting with solution A (deionized water containing 0.12% TFA) and progressing to 60% solution B (acetonitrile containing 0.1% TFA) over a 60 min period, as indicated by the black dashed line. (**A**) Cvill6 was identified in fraction F-II-10 in the peak eluted at 25.5 min (denoted by a green asterisk). (**B**) Cvill7 appeared in fraction F-II-11 eluted at 23.6 min (marked by a red asterisk).

**Figure 2 toxins-17-00279-f002:**
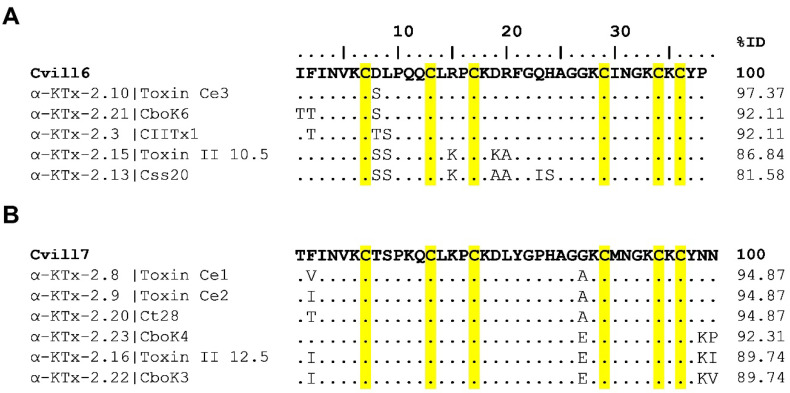
Multiple sequence alignments of Cvill6 and Cvill7 peptides and other scorpion potassium toxins. Peptide toxins closely related (>80% identical) to Cvill6 (**A**) and Cvill7 (**B**) are shown. The systematic names followed by the common names are shown in the left column. %ID indicates percent amino acid identity. Conserved cysteine residues are highlighted in yellow.

**Figure 3 toxins-17-00279-f003:**
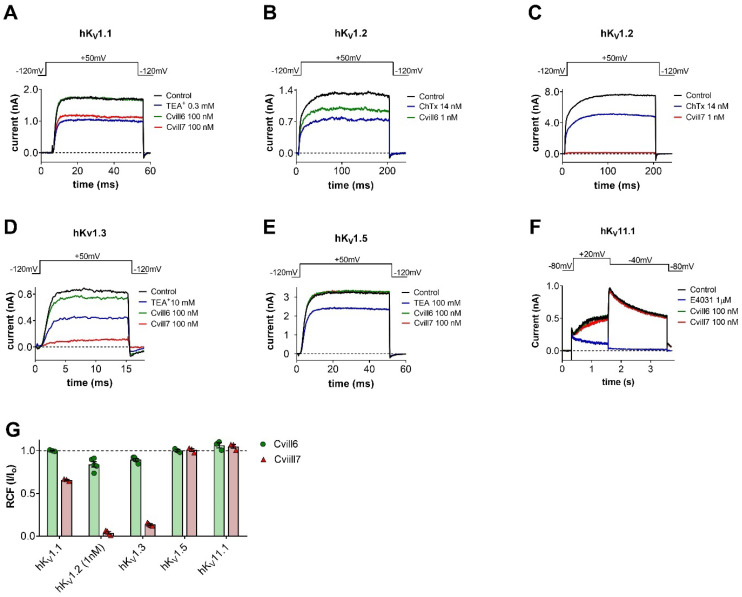
Effect of Cvill6 and Cvill7 on Kv ion channels. (**A**–**F**) Representative current traces shown for each channel were recorded in the absence (control, black) and in the presence of 100 nM (except panel C for Kv1.2, where 1 nM was present) of either toxin (green: Cvill6; red: Cvill7) at steady-state block (**A**–**D**) or after 6–12 depolarization pulses (3 min (**E**,**F**)). Complete solution exchange in the recording chamber was confirmed using TEA*^+^* (tetraethylammonium chloride), ChTx (charybdotoxin), and E4031, known inhibitors of appropriate channels as indicated (blue). Voltage protocols are displayed above the current traces in each panel. For extracellular and intracellular solution composition see Section 5. (**G**) The remaining current fraction (RCF, I/I_0_) values were calculated as the ratio of the peak currents in the presence (I) or absence (I_0_) of 100 nM (or 1 nM for Kv1.2) of Cvill6 and Cvill7 at equilibrium block or after 3 min of toxin application. Bars (green: Cvill6; light red: Cvill7) with individual data points (green solid circles for Cvill6 or red solid triangles for Cvill7) represent the RCF values determined from individual cells. Error bars indicate the mean ± SEM (*n* = 3−6).

**Figure 4 toxins-17-00279-f004:**
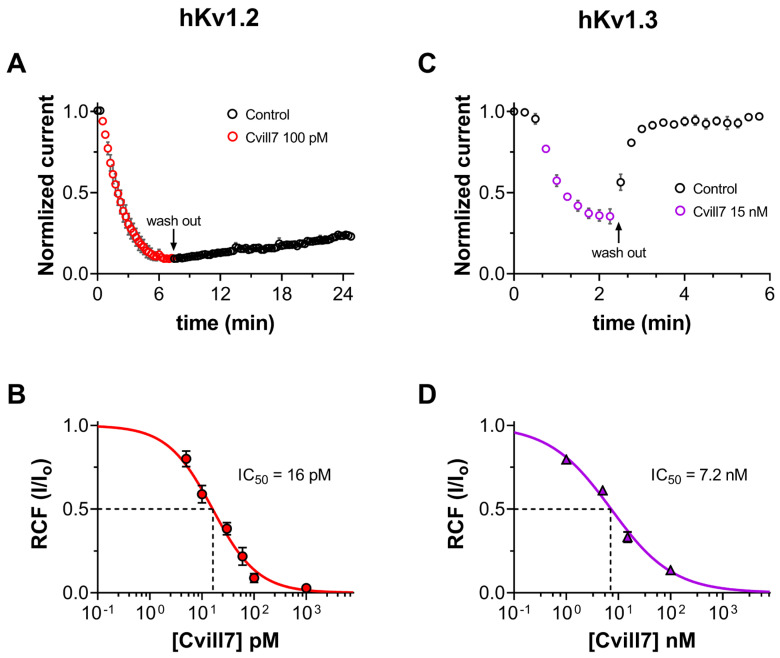
Inhibition of hKv1.2 and hKv1.3 currents by Cvill7. (**A**,**C**) The time course of development and recovery of Kv1.2 (**A**) and Kv1.3 (**C**) current inhibition. Normalized peak (I/I_0_, refer to the text) currents from different cells were averaged at each time point and plotted as a function of time (mean ± SEM, *n* ≥ 3). Data points in red ((**A**), Kv1.2) and in purple ((**C**), Kv1.3) represent the application of 100 pM or 15 nM of Cvill7 to Kv1.2 or Kv1.3 currents, respectively. Upon reaching equilibrium, cells were perfused with bath solution lacking the toxin (arrow, wash-out) to demonstrate the reversibility of block (empty black circles). (**B**,**D**) Concentration-dependent block of Kv1.2 (**B**) and Kv1.3 (**D**) currents by Cvill7. A Hill equation (see Section 5 for details) was fitted to the remaining current fraction (RCF, I/I_0_) values (red solid circles in (**B**), purple solid triangles in (**D**)) calculated at various toxin concentrations (solid lines). The best fits yielded IC_50_ = 16 pM, H = 1.03 for Kv1.2 (**B**) and IC_50_ = 7.2 nM, H = 0.73 for Kv1.3 (**D**). Error bars represent the standard error of the mean (SEM), and *n* = 3–7.

**Figure 5 toxins-17-00279-f005:**
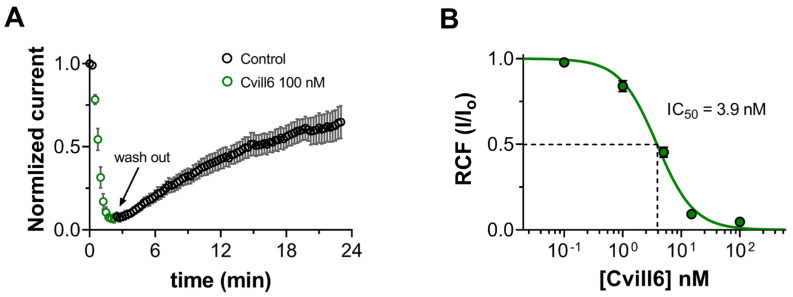
Cvill6 blocks Kv1.2. (**A**) Time course of onset and relief of Kv1.2 inhibition by Cvill6. Data points (averaged normalized currents, see legend in and text for Figure 4 for details) in green indicate the Cvill6 (100 nM) application to bath solution and those in black represent the perfusion of cell with toxin-free solution (wash-out) following the equilibrium block. (**B**) Dose-dependent block of Kv1.2 by Cvill6. Fitting of RCF values (green solid circles) with a Hill equation resulted in an IC_50_ = 3.9 nM and H = 1.4. Error bars denote SEM and *n* = 3−6.

**Figure 6 toxins-17-00279-f006:**
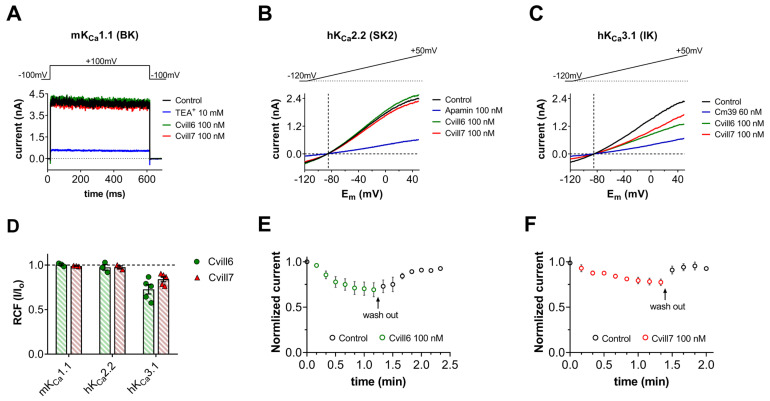
Effect of Cvill toxins on Ca^2+^-activated K^+^ channels. (**A**–**C**) Representative current traces for mKCa1.1 (**A**), hKCa2.2 (**B**), and hKCa3.1(**C**) in the control solution (traces in black) and at equilibrium block or after 12–20 depolarization pulses (~3 min) upon perfusing 100 nM of Cvill6 (green) or Cvill7 (red). Traces in blue indicate the equilibrium block in the presence of positive control TEA^+^, apamin, and Cm39 for the respective channel (for details see Section 5). Voltage protocols are shown in the inset above the representative current traces. (**D**) Summary of RCF values (I/I_0_, where I_0_ is the peak current in control solution, and I is the peak current in the presence of 100 nM of either toxin at steady-state block or after ~3 min of toxin application). For KCa2.2 and KCa3.1, peak currents were measured at +48 mV (time point 148 ms) of the voltage ramp. Bars (green: Cvill6, light red: Cvill7) with data points (green solid circles for Cvill6 or red solid triangles for Cvill7) represent the RCF (I/I_0_) values determined from individual cells. (**E**,**F**) Time course of inhibition of KCa3.1 by Cvill6 (**E**) and Cvill7 (**F**). The averaged values of normalized peak currents (see legend in and text for Figure 5 for details) indicate the application of 100 nM of Cvill6 (green empty circles, (**E**)) or Cvill7 (red empty circles, (**F**)) to the bath solution and black empty circles represent the perfusion of cell with toxin-free solution (wash-out) following the equilibrium block. Error bars indicate the mean ± SEM (*n* = 3−5).

**Table 1 toxins-17-00279-t001:** Kinetic parameters of Cvill toxins and Kv1 ion channels.

Toxins	Channel	k_on_ (nM^−1^s^−1^)	k_off_ (s^−1^)	*n*
Cvill7	Kv1.2	0.1±0.015	$1.7\times 10^{-4}\pm 1.5\times 10^{-5}$	4
	Kv1.3	$1.3\times 10^{-3}\pm 1.6\times 10^{-4}$	$0.032\pm 6.8\times 10^{-3}$	3
Cvill6	Kv1.2	$2.9\times 10^{-4}\pm 2.5\times 10^{-5}$	$9.1\times 10^{-4}\pm 1.6\times 10^{-4}$	5

The k_on_ and k_off_ parameters were calculated from time constants (τ_on_ and τ_off_) of development and relief from the block in the presence of 100 pM Cvill7 (for Kv1.2), 15 nM Cvill7 (for Kv1.3), or 100 nM Cvill6 (for Kv1.2).

## Data Availability

The original contributions presented in this study are included in the article/Appendix A. Further inquiries can be directed to the corresponding authors.

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
