# Peer review of "Cvill6 and Cvill7: Potent and Selective Peptide Blockers of Kv1.2 Ion Channel Isolated from Mexican Scorpion Centruroides villegasi"

_toxins, 2025, doi:10.3390/toxins17060279_

Round 1
Reviewer 1 Report
Comments and Suggestions for Authors
The authors reported the discovery of two new peptides, Cvill6 and Cvill7, isolated from the venom of the scorpion Centruroides villegasi. As their sequence are highly similar to members of the α-KTx subfamily, they investigated the effects of Cvill6 and Cvill7 on voltage-gated potassium channels. Notably, Cvill7 demonstrates selective inhibition of Kv1.2. However, my primary concern lies in the insufficient characterization of the two new peptides. While the main HPLC chromatograms and aa sequence are included, the identification of novel toxins from natural sources necessitates additional analytical validation. In particular, the MS figure and NMR data should be provided, which are essential to unambiguously verify the identity and purity of the peptides. Without these critical datasets, the claims of discovering new toxins remain inadequately supported.
My comments in details:
- A significant portion of method description have been inappropriately placed in the Results section. Like in 2.1 “The purification of Centruroides villegasi …In brief, the venom was extracted from scorpions via electric stimulation, dissolved in sterile water, and centrifuged at 15,000 rpm at 4°C for 15 minutes. The supernatant was collected, lyophilized, and stored at -20°C. …” These descriptions should be moved to section 5. Materials and Methods.
- The HPLC traces corresponding to all three steps of the purification process must be included, either in the main text or as Supplementary Materials. Additionally, MS figures validating the molecular weights of the purified peptides should be provided in 2.1 or as Supplementary Materials.
- In 2.2, Alphafold3 could be used to predict the 3D structure of a new peptide, however, experimental NMR data remains mandatory to verify structural details and confirm purity. Specifically, 1H one-dimensional NMR data as well as 1H-1H two-dimensional TOCSY/ NOESY data should be supplemented. If natural yields are insufficient, synthetic peptides should be used to acquire these datasets.
- In figure 5 C&D the X axis are not clearly defined. Additionally, IC50 values for both Cvill6 and Cvill7 targeting Kv2 and Kv1.3 channels should be determined to comprehensively characterize their potency and selectivity.
Author Response
First of all, we would like to thank Reviewer 1 for the time and efforts devoted to reviewing the manuscript. We are also grateful for the comments and the recommendations. Please find our itemized responses below along with the critiques:
- A significant portion of method description have been inappropriately placed in the Results section. Like in 2.1 “The purification of Centruroides villegasi …In brief, the venom was extracted from scorpions via electric stimulation, dissolved in sterile water, and centrifuged at 15,000 rpm at 4°C for 15 minutes. The supernatant was collected, lyophilized, and stored at -20°C. …” These descriptions should be moved to section 5. Materials and Methods.
[Answer 1]: We thank the reviewer for noticing this issue. We moved the methodology related sentences from results sections to “Materials and Methods section, and we have described the methodology in detail about purification of toxins from the venom.
- The HPLC traces corresponding to all three steps of the purification process must be included, either in the main text or as Supplementary Materials. Additionally, MS figures validating the molecular weights of the purified peptides should be provided in 2.1 or as Supplementary Materials.
[Answer 2]: Thank you very much for the comment. We previously published a descriptive methodology (including all three steps with chromatograms) for purifying Na⁺ channel-targeting toxins from the Centruroides villegasi scorpion. The current study, which focuses more on K⁺ channel toxins, is directly linked to that earlier work. We have now included the missing purification set chromatograms as the Supplementary Materials. We have also included the MS data as supplemental figure. We are sorry for overlooking this important data and not providing the necessary information for the readers.
- In 2.2, Alphafold3 could be used to predict the 3D structure of a new peptide, however, experimental NMR data remains mandatory to verify structural details and confirm purity. Specifically, 1H one-dimensional NMR data as well as 1H-1H two-dimensional TOCSY/ NOESY data should be supplemented. If natural yields are insufficient, synthetic peptides should be used to acquire these datasets.
[Answer 3]: We completely agree with the reviewer that firm structural data can only be obtained using NMR. Our team also follows this principle, and we determined the structure of the peptides experimentally. Unfortunately, in this case the limited amount of the biological material prevented the determination of the structure by NMR. As for determining the structure of the synthetic/recombinant peptide. We do this also routinely as well and use the NMR data to verify that the synthetic/recombinant peptides are identical and as such we can use the synthetic peptide for pharmacology. This, however, requires the knowledge of 3D structure of the native peptide, which we do not have in this case.
To reflect that the structure in the manuscript is a model, we emphasized this fact more clearly in the manuscript. Moreover, we have aligned the structure of MgTx (NMR structure available from alpha-KTx family 2) with the AF3 models of the novel peptides to show the structural feature similarity. This was suggested by Reviewer 2 as a solution for strengthening the structure prediction.
4. In figure 5 C&D the X axis are not clearly defined. Additionally, IC50 values for both Cvill6 and Cvill7 targeting Kv2 and Kv1.3 channels should be determined to comprehensively characterize their potency and selectivity.
[Answer 4]: As per our understanding the X axis is clearly labeled in all panels of figure 5. We indicated the concertation of the peptides using brackets and the unit of the concentration is also displayed (nM or pM). We have also completed the concentration-response curves in Fig 5B (Cvill7 for Kv1.2), Fig 5 D (Cvill7 for Kv1.3) and Fig 6 A (Cvill6 for Kv1.2) wherever the concentrations of the peptides were reasonable considering the availability of the native peptide. We could not obtain the concentration-response for the inhibition of Kv1.3 by Cvill6, this would have required concentrations up 10-50 to µM based on the IC50 estimate of ~0.84 µM. Based on the same argument we have estimated the IC50 of Cvill7 for of Kv1.1(~192 nM).
Finally, we would like to thank Reviewer 1 again for the valuable suggestions. We hope that Reviewer 1 will find the extensively revised manuscript suitable for publication in Toxins.
On behalf of all authors,
Corresponding authors
Reviewer 2 Report
Comments and Suggestions for Authors
This study presents a well-executed investigation into the isolation and characterization of two novel scorpion toxins. The identification of Cvill7 as a highly potent and selective inhibitor of the Kv1.2 channel represents a significant advancement with potential therapeutic implications. The manuscript is generally well-written, and the results are clearly presented and discussed within a relevant scientific framework. The detailed methodology enhances the reproducibility of the findings. neverthless I have some comments to improve the mansucript.
-
The authors state that "Since fraction II is known to contain peptides that inhibit ion channels…". What evidence or previous studies support this claim? A reference or a brief explanation would help justify this statement.
-
The rationale for selecting only F-II-10 and F-II-11 from fraction II is not sufficiently explained. Please clarify the criteria used for this selection.
-
The authors should provide experimental data demonstrating the purity and molecular weight of F-II-10 and F-II-11, to confirm that they contain single, well-defined peptide species.
-
There is an inconsistency in the terminology: the main text refers to "similarity" while Figure 2's legend mentions "identity". These are distinct parameters in sequence comparison and should not be used interchangeably. Please clarify and correct throughout the manuscript.
-
While structural alignment using NMR structures of homologous peptides is useful, it does not validate the predicted structure of Cvill7 or Cvill6. The authors should include structural alignments with known toxins such as MgTx and/or HgTx for more robust comparative analysis.
-
CHO cells are known to express endogenous potassium currents. How did the authors account for this background activity when calculating the relative current fraction (RCF)? This concern is also relevant to Figure 5, where the authors state that "loss of current apparently saturated to 95% of the initial". Could the remaining 5% reflect endogenous currents?
-
In Figure 5, panels A and C: do these represent recordings from a single cell or are they averaged traces? If individual traces, averaged data with error bars should be provided to reflect biological variability.
-
In Figure 5, panels B and D should be displayed using the same scale to allow direct comparison of binding affinities.
-
Consider inverting Figures 4 and 5. Presenting the affinity data first would better support the subsequent analysis of selectivity.
-
Please fit the association and dissociation kinetics with appropriate exponential models and report the corresponding rate constants. This would provide more quantitative insight into binding kinetics.
-
Quantification of current inhibition and recovery for Kv1.2 and Kv1.3 should be based on multiple cells. One representative example (as shown in Figure 5 A and C) is insufficient. The same applies to Figure 6.
-
The authors mention taking into account the “extremely slow blocking kinetics of Cvill7 for Kv1.2” and that they “applied the toxin for a sufficient period to reach the complete apparent equilibrium block.” This is a sound methodological choice. However, please provide time-course data of current inhibition to substantiate this statement, especially for Kv1.2 (Figure 5) and Kv1.3 (Figure 6).
-
The fit of concentration-response curves provides IC₅₀ values, not Kd. Please clarify this distinction in the manuscript.
-
In Figure 7, were the four conditions for each channel recorded from the same cell? This information should be clearly stated. It would also be valuable to include time-course data for toxin effects, particularly for the KCa3.1 channel, along with averaged results ± SEM.
Round 2
Reviewer 1 Report
Comments and Suggestions for Authors
The authors acknowledged that structural determination of peptides, particularly novel ones, requires NMR data. However, they declined to perform peptide synthesis and NMR structural determination, stating that “it requires the knowledge of 3D structure of the native peptide”, which is confusing. Since they already have the amino acid sequence and cysteine framework of the native peptide, what more is needed for peptide synthesis? They can simply co-elute the native peptide with the synthetic peptide to confirm the peptide folds. It is inappropriate to claim that “The primary structures of these peptides were determined using Edman degradation and compared with known potassium scorpion toxins (KTxs)” Edman degradation only provides amino acid sequence information, and in many cases, it provides tentative sequences. Additionally, the MS data for both peptides do not appear correct. It is highly likely that the two native peptides are not pure. Contaminants could compromise functional assays, and therefore, pharmacological characterization of synthetic peptides should be required to confirm their activity. Without synthesis, structural validation, and purity verification, the structural and functional conclusions remain incomplete.
Author Response
Responses to reviewer 1 round 2
First of all, we would like to thank Reviewer 2 for the time and efforts devoted to reviewing the manuscript. Please find our itemized responses below along with the critiques:
- The authors acknowledged that structural determination of peptides, particularly novel ones, requires NMR data. However, they declined to perform peptide synthesis and NMR structural determination, stating that “it requires the knowledge of 3D structure of the native peptide”, which is confusing. Since they already have the amino acid sequence and cysteine framework of the native peptide, what more is needed for peptide synthesis? They can simply co-elute the native peptide with the synthetic peptide to confirm the peptide folds.
To our understanding, we made it clear in the manuscript that we do not have the cysteine pairing; what we showed in the last submission was a predicted Cys pairing pattern. Determination of the Cys pairing, recombinant or synthetic production, validation and structure determination seem to be too far-reaching considering the aim of the publication. We are characterizing the basic pharmacology of the peptides. We agree that for in-depth pharmacological and structure-function relationship studies, we need the 3D structure. We are planning to do this, but we think that that should be the subject of a different paper, not the current one.
However, in response to the critiques of Reviewer 1 we moved the predicted tertiary structures into the supplemental material to avoid potential confusion of the readers or misinterpretation of the figure.
- It is inappropriate to claim that “The primary structures of these peptides were determined using Edman degradation and compared with known potassium scorpion toxins (KTxs)” Edman degradation only provides amino acid sequence information, and in many cases, it provides tentative sequences.
This comment about using the term “primary structure” is very confusing for us.
As the review highlights that “Edman degradation only provides amino acid sequence information”. According to the defined level of protein organization, the amino acid sequence constitutes the primary structure. We used this term for the amino acid sequences of Cvill peptides because we got the complete and accurate sequences using direct automated Edman degradation. We agree that in the absence of this, we cannot claim a primary structure. The theoretical molecular weights (MW) based on the determined amino acid sequences of pure peptides are identical to the determined MW of the pure fractions using MS (Fig. 2 for primary sequences and Figure S4 for MS), which confirms the completeness and accuracy of amino acid sequencing.
We have used this term to refer amino acid sequence of scorpion peptides determined using Edman degradation in several previous publications, some examples are here [1-7].
- Additionally, the MS data for both peptides do not appear correct. It is highly likely that the two native peptides are not pure. Contaminants could compromise functional assays, and therefore, pharmacological characterization of synthetic peptides should be required to confirm their activity. Without synthesis, structural validation, and purity verification, the structural and functional conclusions remain incomplete.
In response to the criticism we repeated the MS analysis of both peptides using a high-resolution mass spectrometer (methodology described in the revised version of the manuscript) to reconfirm the molecular weights (MW) of native peptides and purity. As shown below (and in revised supplementary data) the MW reported by our previous MS data are identical to the new data. In MS, we observed a single component in each sample having the corresponding MW, demonstrating the high purity of native peptides.
Moreover, we argue that the native peptides are substantially pure:
- In the RP-HPLC chromatograms of 3rd step of purification scheme of venom as shown in Figure 1, well-isolated sharp peaks appeared for both peptides, rendering negligible chances of contamination by other peptides; Cvill6: peak indicated with green asterisk at time point of 25.5 min (Figure 1A), Cvill7: peak indicated with red asterisk at time point of 23.6 min (Figure 1B).
- Further re-analysis of these fractions using RP-HPLC showed a single sharp peak for both peptides at the corresponding retention time points (Figure S2A and B).
- Theoretical (calculated) MWs based on the determined amino acid sequences of purified fractions (F-II-10 25.5 and F-II-11 23.6) using the automated Edman degradation are in full agreement with the MS data. This further verifies that the purified fractions contain a single kind of component with the corresponding MW.
- In new MS data of the same purified fractions (F-II-10 25.5 and F-II-11 23.6), we observed only a single kind of component with corresponding mass of Cvill peptides, which also confirm the high purity level of native peptides.
Together, all these observations verify the purity of native peptides and thus, functional characterizations are not influenced by impurities.
Using the same purification scheme, we have reported several peptides with functional characterization previously [2, 4, 6, 8, 9]. Many of these were synthesized or recombinantly produced with identical functional activities in subsequent studies, confirming that this 3-step purification scheme is very suitable to isolate the pure peptides from soluble venom (see e.g. Anuroctoxin [3, 10]).
We sincerely hope that Reviewer 1 will consider our arguments above positively and reconsider the recommendation and will accept the R2 version of the manuscript.
On behalf of all authors:
Corresponding authors
References:
- Gurrola, G.B., et al., Structure, function, and chemical synthesis of Vaejovis mexicanus peptide 24: a novel potent blocker of Kv1. 3 potassium channels of human T lymphocytes. Biochemistry, 2012. 51(19): p. 4049-4061.
- Naseem, M.U., et al., Characterization and chemical synthesis of Cm39 (α-KTx 4.8): a scorpion toxin that inhibits voltage-gated K+ channel KV1. 2 and small-and intermediate-conductance Ca2+-Activated K+ channels KCa2. 2 and KCa3. 1. Toxins, 2023. 15(1): p. 41.
- Bagdáany, M., et al., Anuroctoxin, a new scorpion toxin of the α-KTx 6 subfamily, is highly selective for Kv1. 3 over IKCa1 ion channels of human T lymphocytes. Molecular pharmacology, 2005. 67(4): p. 1034-1044.
- Riaño-Umbarila, L., et al., Toxic Peptides from the Mexican Scorpion Centruroides villegasi: Chemical Structure and Evaluation of Recognition by Human Single-Chain Antibodies. Toxins, 2024. 16(7): p. 301.
- Restano-Cassulini, R., et al., Characterization of Sodium Channel Peptides Obtained from the Venom of the Scorpion Centruroides bonito. Toxins, 2024. 16(3): p. 125.
- Corzo, G., et al., A selective blocker of Kv1. 2 and Kv1. 3 potassium channels from the venom of the scorpion Centruroides suffusus suffusus. biochemical pharmacology, 2008. 76(9): p. 1142-1154.
- Olamendi-Portugal, T., et al., Isolation, chemical and functional characterization of several new K+-channel blocking peptides from the venom of the scorpion Centruroides tecomanus. Toxicon, 2016. 115: p. 1-12.
- Shakeel, K., et al., Of Seven New K+ Channel Inhibitor Peptides of Centruroides Bonito, α-KTx 2.24 Has a Picomolar Affinity for Kv1. 2. Toxins, 2023. 15(8): p. 506.
- Beltrán-Vidal, J., et al., Colombian scorpion Centruroides margaritatus: Purification and characterization of a gamma potassium toxin with full-block activity on the hERG1 channel. Toxins, 2021. 13(6): p. 407.
- Borrego, J., et al., Recombinant expression in Pichia pastoris system of three potent Kv1. 3 channel blockers: Vm24, Anuroctoxin, and Ts6. Journal of Fungi, 2022. 8(11): p. 1215.
Reviewer 2 Report
Comments and Suggestions for Authors
The authors have adressed all comments. I have no more comments
Author Response
Dear Reviewer 2,
Thank you very much for recommending the acceptance of our manuscript.
Sincerely
Round 3
Reviewer 1 Report
Comments and Suggestions for Authors
I would assume six cysteines in a toxin suggest the potential for an ICK motif, and a simple CD spectrum could be used to compare the secondary structure of the native and synthetic peptides. I have made it clear that NMR is mandatory for the claim of new toxin discovery, and I will not endorse the publication of this current manuscript without it.